# Effects of Multi-Pass Friction Stir Processing on Microstructures and Mechanical Properties of the 1060Al/Q235 Composite Plate

**Jian Wang [1,*], Yun Cheng [1,2], Bo Li [3] and Cheng Chen [1]**

[1] School of Mechanical and Power Engineering, Nanjing Tech University, Nanjing 211816, China; chengy1997@163.com (Y.C.); chencheng2017@njtech.edu.cn (C.C.)

[2] School of Materials Science and Engineering, Jiangsu University of Science and Technology, Zhenjiang 212003, China

[3] Additive Manufacturing and Intelligent Equipment Research Institute, School of Mechanical and Power Engineering, East China University of Science and Technology, Shanghai 200237, China; libo@ecust.edu.cn

[*] Correspondence: njjwang@njtech.edu.cn; Tel.: +86-255-813-9352

**Abstract:** Steel cuttings, holes and cracks always exist at the interfaces in the composite plate. Multi-pass friction stir processing (M-FSP) is proposed in this research to optimize the interface microstructure and the interface connection for the 1060Al/Q235 composite plate. Results show that the microstructures of 1060Al after M-FSP are fine and uniform owing to the strong stirring effect and recrystallization. Micro-defects formed by the welding can be repaired by the M-FSP. However, tunnel defects can also be formed in the matrix of aluminum by M-FSP, especially when the passes are one and two. The melting block and the melting lump in the composite plates are easy to become the source of crack. The shear strengths and the bending properties for the 1060Al/Q235 composite plate after M-FSP are the best when the passes are three, with the tool rotation speed of 1200 rpm and the forward speed of 60 mm/min. The optimized interfaces for the composite plate after M-FSP are mainly by the metallurgical bondings, with a certain thickness and discontinuous mechanical connections. Therefore, the crack extension stress is the largest and the mechanical properties are the best.

**Keywords:** friction stir processing; aluminum/steel composite plate; multi-pass; bonding interface; mechanical properties

## 1. Introduction

Aluminum/steel composite plate is well known as a kind of common lightweight composite material and has been widely used in automobile, ship, aerospace and pressure vessel fields [1–3]. Compared with the single metallic materials, the aluminum/steel composite plate not only has superior properties but also greatly reduces the weight of equipment. However, the different physical and chemical properties of aluminum and steel make them unable to form effective and strong joint by fusion welding.

Explosive welding is one of the joining methods belonging to the solid state welding process. The metal plates are joined together under a very high pressure and there is local plastic deformation at the interface. The interfaces after explosive welding are almost metallurgical bondings and even stronger than the base metals. Similar and dissimilar materials can be joined by the explosive welding [4]. However, steel cuttings, holes, cracks and other defects caused by the high temperature and the excessive explosion pressure restrict the application of composite plates [5–7].

Friction stir processing (FSP) developed from friction stir welding (FSW) has been invested in improving the performance of materials [8–10]. Gupta et al. found that FSP can refine grain structures and eliminate defects like porosity in the matrix of AZ31 magnesium alloy [11]. Barmouz found that FSP can refine the second particle's size and enhance the dispersion distribution of the second particles for Cu/SiC composites [12]. Nakata found that FSP can improve the hardness and tensile strength for ADC12 aluminum die casting alloy by grain refinement and disappearance of cold flake [13]. Pourali et al. investigated the different tool rotations and weld speeds for lap weld joint of 1100Al/St37 steel composite plate. They found that the tensile strength of composite with high tool rotation and low weld speed is superior since the interface of this joint formed a properly thick bonding layer compared with other parameters [14].

However, FSP can also reduce the mechanical properties when the parameters are inappropriate. Gupta et al. also found that tunnel defects are both formed in AZ31 magnesium alloy whether in the truncated conical tool or the cylindrical tool [11]. Kima et al. found that tunnel defects are easily formed in ADC12 aluminum die casting alloy during welding [15].

In order to improve the mechanical properties of composite plate, obtaining the appropriate interface is the main method since the interface of composite plate is the main carrying structure during service. Thus, there are two main factors that affecting the microstructures and mechanical properties of composite plate after FSP with different passes, which include the interface connections and the tunnel defects.

In the first part, the bonding strength of composite plate is related to the interface connection mechanisms, which include the mechanical connection and the metallurgical bonding. Pourali et al. found that mechanical connections are the main connection methods of aluminum/steel composite plate under the condition of low welding speed [14]. During explosive welding, the energies of explosion flow cause some of the steel and aluminum soften or even melt, mix and finally form the morphologies of hooks or vortexes at the interface [16,17]. Meanwhile, defects such as steel cuttings and microcracks caused by explosion weaken the strength of composite. Since FSP can refines the second particles size and reduces the cold flake of aluminum die casting alloy [12,13], the steel cuttings and microcracks could also be repaired by strong stirring of aluminum. Therefore, FSP is proposed to repair the defects formed by explosive welding.

The metallurgical bondings mainly include intermetallic compounds (IMCs) and barely have few solid solutions. In the process of welding, the formation mechanisms of interface layer are diffusion and metallurgical reaction in the result of temperature and pressure [18,19]. Since the mutual solubility of aluminum and steel is few, the mechanical properties are poor even if forming the solid solution. Therefore, IMCs are interface connections basis for the composite plate. $Fe_3Al$, $FeAl$, $Fe_2Al_5$, $FeAl_2$ and $FeAl_3$ ($Fe_4Al_{13}$) are the dominating forming IMCs during welding process. According to the content ratio of aluminum to steel, IMCs can be divided into two categories—IMCs rich in Fe which are rigid and IMCs rich in Al which are toughness [20–24]. In addition, Bozzi et al. found that when the thickness of IMCs is 8 μm, mechanical properties of the composite plate are the best. While the thickness increases to 42 μm, mechanical properties of the composite plate are the worst [25]. In other words, the thickness of IMCs for the composite plate is a key factor influencing its mechanical properties. When the thickness of IMCs is lower than a certain value, the mechanical properties of composite plate improve with the IMCs thickness increasing [26,27]. With the FSP passes increasing, the accumulated heat input increases and finally IMCs thickness is also getting thicker. However, too thick IMCs decrease the bonding strength of interface. Thus, it is necessary to use M-FSP for obtaining the appropriate thickness of IMCs.

In the second part, tunnel defects are usually found in the composite plate after the single-pass FSP [11,15,16,28]. Kima investigated different tool plunge downforces of FSW for ADC12 aluminum die casting alloy. They found that tunnel defects are caused by the insufficient heat input [15]. Javad et al. presented a mathematical model for the heat input generation during friction stir welding of 1060 aluminum alloy. They also found that the insufficient heat input can form tunnel defects in the matrix of aluminum during welding [29]. Due to the insufficient heat input, the insufficient flow of material stirred causes the tunnel defects. Therefore, the heat input is increased by increasing the

number of stirring passes to eliminate the tunnel defects. Meanwhile, the plastic flow of aluminum can also increase with the increase of passes. Thus, M-FSP is used to repair the tunnel defects in the composite plate.

In order to find the relationships between microstructures and mechanical properties of aluminum/steel composite plate after FSP with different passes, the 1060Al/Q235 composite plates are chosen as the objects to study on account of its superior plasticity and extensive application. Meanwhile, compared to other alloys, the 1060Al and the Q235 have fewer alloying components than other materials, which makes it easy to study the interface fracture models. The optimized pass of FSP is obtained by microstructures observation and mechanical properties investigation. The above research provided a good data basis in the production of aluminum/steel composite plate in the future.

## 2. Experimental Materials and Methods

The initial material for the sample used in this experiment was the 1060Al/Q235 composite plate and its chemical compositions (wt.%) were measured by ICP direct reading spectrometer, as shown in Table 1. It conformed to 1060Al and Q235 standard specification. The thickness of aluminum/steel composite plate was 6 mm and the thickness of aluminum and steel was both 3 mm.

**Table 1.** Chemical compositions for 1060Al/Q235 composite plate (unit: wt.%).

| Ele. | Fe | Al | Si | Mn | Ti | Zn | C | S | P |
|------|------|-------|------|------|-------|-------|------|-------|-------|
| 1060Al | 0.19 | Bal. | 0.15 | 0.03 | 0.017 | 0.012 | - | - | - |
| Q235 | Bal. | 0.029 | 0.12 | 0.32 | - | - | 0.13 | 0.009 | 0.022 |

M-FSP was performed using a FSW machine modified from a CNC Vertical milling machine (XK7132, Goldentop Company, Nanjing, China) with a tool rotation speed of 1200 rpm and forward speed of 60 mm/min according to the previous study. The axial force during M-FSP process was 10 kN. The FSW tool was made of W-Re alloy and the FSW tool geometry consisted of a threaded conical pin, with three flats (7 mm diameter) and a spiral (scroll) shoulder with a diameter of 25 mm. The tilt angle for the FSW tool during welding was 2°. The pin tool plunge depth was 2.5 mm. The stirring head was inserted in the aluminum. The rotational orientation of the stirring head was counterclockwise. After each pass, the composite plate was cooled down about 7 min to avoid the influence residual heat from past passes and then the subsequent pass was performed. The overlap rate was 100%.

The sample was polished until the surface was smooth and scratch-free. Then, the microstructures of 1060Al after M-FSP were etched with Keller's reagent to imaging. The microstructures of Q235 were etched with 4% nitric acid alcohol prior to imaging. The microstructures were analyzed by an optical microscope (OM, DM2700, Leica Company, Wetzlar, Germany). A scanning electron microscope (SEM, Quanta 450, FEI Company, Hillsboro, USA) was used to observe the fracture surface of shear specimens. A transmission electron microscope (TEM, Tecnai T12, FEI Company, Hillsboro, USA) was used to observe structures of the interface.

Based on the GB/T2651-2008 and the GB/T232-2010 standards, the shear specimens and the bending specimens were designed, as shown in Figure 1. Three shear testing samples and two bending testing samples were tested for each group in order to give a better reliability for the testing datum.

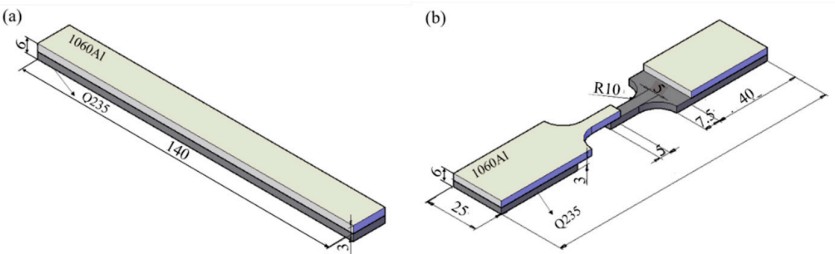

**Figure 1.** Dimensions of mechanical samples: (**a**) bending specimen and (**b**) shear specimen (unit: mm).

The test was carried out using an electronic universal testing machine CSS-44100, at an initial strain rate with a loading speed of $1.2 \times 10^{-4}$/s. The microhardness tests were carried out on HX-1000TM/LCD semi-automatic micro-indentation hardness testing system according to the national standard GB/T 229-2007.

## 3. Results and Discussion

### 3.1. Macro- & Micro- Structures

Figure 2 shows the 1060Al/Q235 composite plate after M-FSP with different passes. The repair zone for the 1060Al/Q235 composite plate after M-FSP with different passes are well-formed with no defects in it. However, with the increase of pass, especially for the passes are four, there shows obvious toe flashes after friction stir processing, as shown in Figure 2e with the red arrows. Beside this, according to the continuous axial force resulting in the toe flashes after M-FSP, the thickness of the aluminum plate becomes thinner gradually. Therefore, only four passes friction stir processing have been applied in this study.

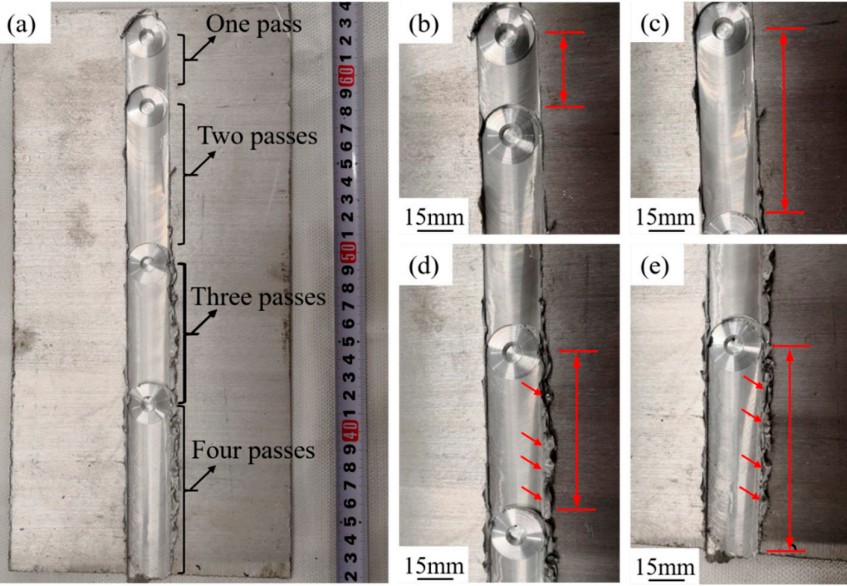

**Figure 2.** 1060Al/Q235 composite plate after multi-pass friction stir processing (M-FSP) with different passes: (**a**) 1060Al/Q235 composite plate after M-FSP; (**b**) single pass; (**c**) two passes; (**d**) three passes; (**e**) four passes.

Figure 3 shows low-magnification OM images of the repair zone for the composite plate after M-FSP. Holes can be observed in the repair zone. The repair zone appears a shape of basins. The right place of the stirring zone (SZ) is the advancing side (AS) while the left place is the retreating side (RS). The AS presents an abrupt "zigzag line" boundary distinctly and the RS is in a shape of a gradual

change. This is because the temperature and flow fields on the AS are different from those on the RS [30–33].

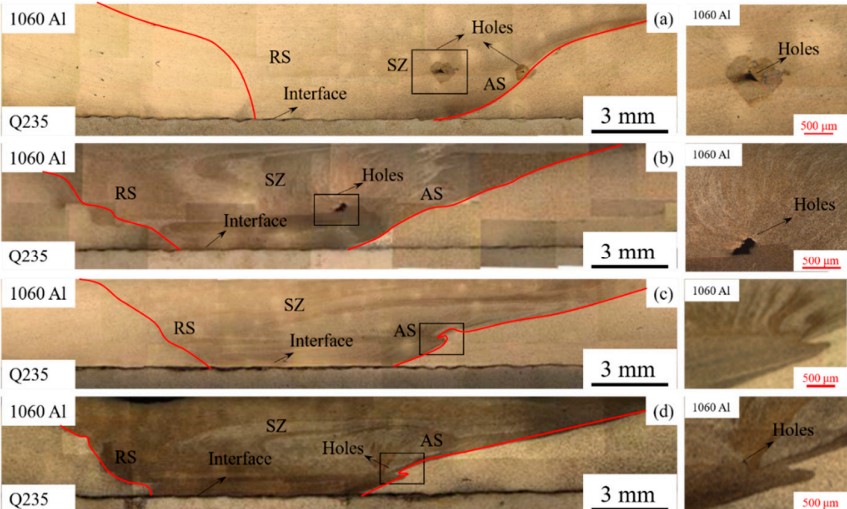

**Figure 3.** Microstructures of the repair zone for 1060Al/Q235 composite plate after M-FSP with different passes: (**a**) single pass; (**b**) two passes; (**c**) three passes; (**d**) four passes.

**Table 2.** Quantities and area of holes in the repair zone for 1060Al/Q235 composite plate after FSP with different passes.

| Pass | Quantities of holes | Total area/$\mu m^2$ |
|:---:|:---:|:---:|
| 1 | 2 | $13.2 \times 10^3$ |
| 2 | 1 | $12 \times 10^3$ |
| 3 | 0 | 0 |
| 4 | 1 | $1.1 \times 10^3$ |

Meanwhile, holes are clearly seen in Figure 3. For the single-pass FSP (Figure 3a), the area of the large hole is $9.8 \times 10^3$ $\mu m^2$ and the smaller one is $3.4 \times 10^3$ $\mu m^2$. However, just one single hole is found in Figure 3b with the area of $12 \times 10^3$ $\mu m^2$. Although the area of the single hole increases, the total areas of the holes decrease. It is confirmed that the numbers and areas of the holes have been repaired by the plastic flow of aluminum after M-FSP. The hole disappears by the influence of repair for the plastic flow in aluminum when the passes of M-FSP are three, as shown in Figure 3c. However, when the passes of M-FSP are four, the hole appears again in the AS area (Figure 3d). Since the distances of these holes from the interface are nearly the same, these holes are the tunnel defects resulted after FSP. The areas of tunnel defects in repair zone with different passes are shown in Table 2. Therefore, with the increase of passes, the tunnel defects decrease. It can be concluded that M-FSP can also repair tunnel defects created by single-pass M-FSP.

Figure 4 shows microstructures of the 1060Al/Q235 composite plate after FSP with different passes. The microstructures of base metal (BM) for the 1060Al are $\alpha$-Al (Figure 4b). The microstructures of BM for Q235 are ferrites and pearlites (Figure 4c). Compared with BM for 1060Al, the grains of repair zone for 1060Al are refined by the stirring effects of M-FSP (Figure 4d). The grain size of 1060Al after M-FSP is about 3.6 $\mu m$, much smaller than that before M-FSP (22.6 $\mu m$). Meanwhile, with the influence of temperature and stir, the microstructures of thermo-mechanical affected zone (TMAZ) are stretched (Figure 4e). Figure 4f shows the interface for the 1060Al/Q235 composite plate after FSP and also exhibits the microstructure of heat affected zone (HAZ) in the Q235 side. Moreover, Figure 4g shows TEM images of interface after M-FSP. The grain size of Q235

is about 600 nm. Even though it only stirs in the aluminum size, the grains for Q235 have been refined by M-FSP, which improves the bonding properties of interface significantly

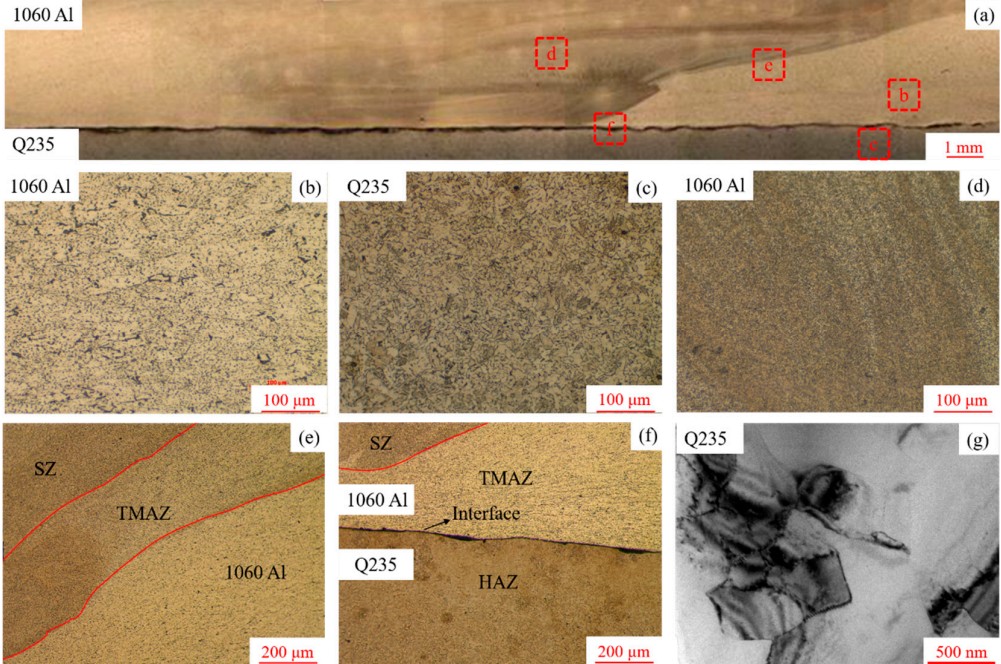

**Figure 4.** Microstructures of composite plate after friction stir processing (FSP) with different passes: (**a**) repair zone; (**b**) base metal (BM) of aluminum; (**c**) BM of steel; (**d**) stirring zone (SZ); (**e**) thermo-mechanical affected zone (TMAZ) in the aluminum side; (**f**) heat affected zone (HAZ) in the steel side and (**g**) transmission electron microscopy (TEM) image near the interface.

### 3.2. Interface Characterization

Figure 5 shows comparisons between the unrepaired interface and the repaired interface. The steel cuttings are clearly seen at the interface of composite plate without FSP (Figure 5a). The sizes of steel cuttings are about 45 μm (Figure 5b). Such a large steel cutting has a severely disadvantage effect on the strength of the interface. Non-uniform and instantaneous temperatures caused by explosive welding are responsible for these steel cuttings. However, these steel cuttings disappear in the interface with FSP (Figure 5c). Through the temperature and plastic flows of aluminum stirred after FSP, these residual steel cuttings react with aluminum and other materials to form new IMCs at the interface. Figure 5d,e show the SEM and Energy Dispersive Spectroscopy (EDS) analysis of the interface for the BM. The black particles are the oxides of aluminum and iron by melting and reacting with oxygen in the air during explosive welding. They are a kind of brittle and hard phase which will reduce the bonding strength for the composite plate. While in Figure 5f,g, there are no steel cuttings and black particles at the interface. The IMCs at the interface are a kind of Al-rich phase [21]. This is to say that FSP can repair the defects remained by explosive welding and optimize microstructures and properties of composite plate.

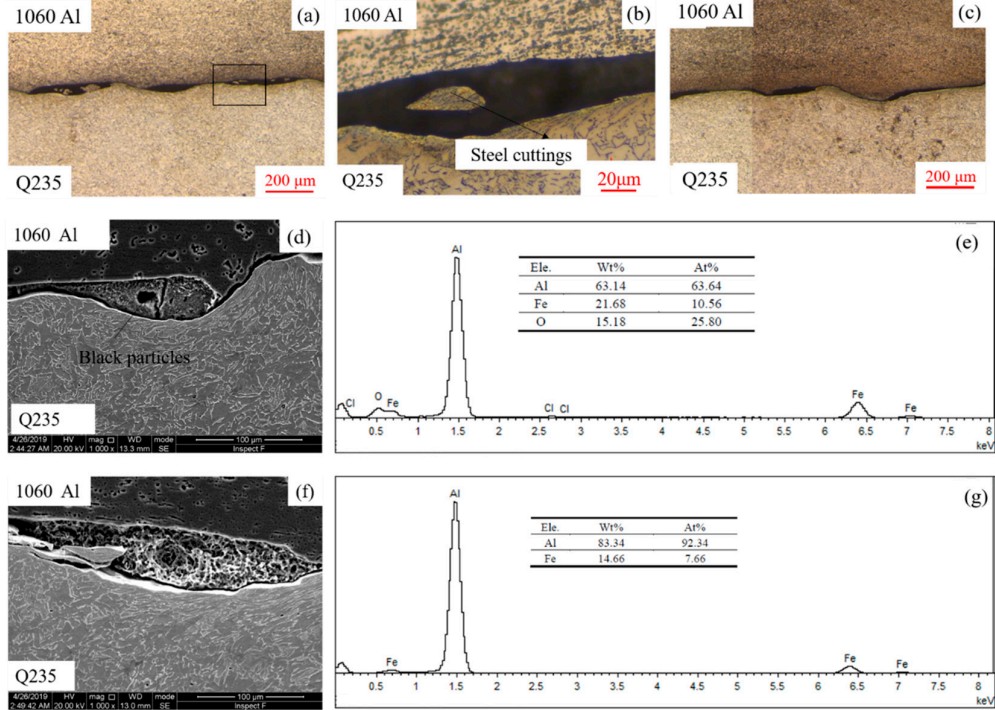

**Figure 5.** Comparisons between the unrepaired interface and the repaired interface: (**a**) and (**b**) unrepaired interface; (**c**) repaired interface; (**d**) and (**e**) Scanning Electron Microscope (SEM) and Energy Dispersive Spectroscopy (EDS) analysis of the unrepaired interface; (**f**) and (**g**) SEM and EDS analysis of the repaired interface.

The interface connections mechanism of composite plate is shown in Figure 6. The mechanical connections consist of hook connections and vortex connections formed by multiple hooks (Figure 6a,b). These mechanical connections are the main connections of interface. Meanwhile, the IMCs formed by reaction of aluminum and steel are the main components of interface during explosive welding. The IMCs on the interface are the basis for the metallurgical bondings, as shown in Figure 6c. Actually, metallurgical bondings are made up of IMCs layers. However, these IMCs separated from the interface are disadvantageous to the properties of composite plate. IMCs show the appearance of melting block if the areas of IMCs are small. Conversely, IMCs show the appearance of melting lump if the areas of IMCs are large. Both the melting block and the melting lump are easy to become originals of crack. The IMCs layer with a certain thickness has a superior effect on the properties of composite plate [22]. Since explosive welding is an instantaneous welding process, the composite plate could not obtain enough thickness IMCs layer. Therefore, M-FSP is used to repair the bonding interface and obtain a proper thickness IMCs layer. With the increase of passes, the proportions of metallurgical bondings increase in the interface ((Figure 6d). The increase of passes brings the increase of heat input, which will aggravate the diffusion and metallurgical reaction at the interface. The IMCs layer becomes thicker due to this. Ultimately, the thick IMCs layer leads to the increase of metallurgical bondings.

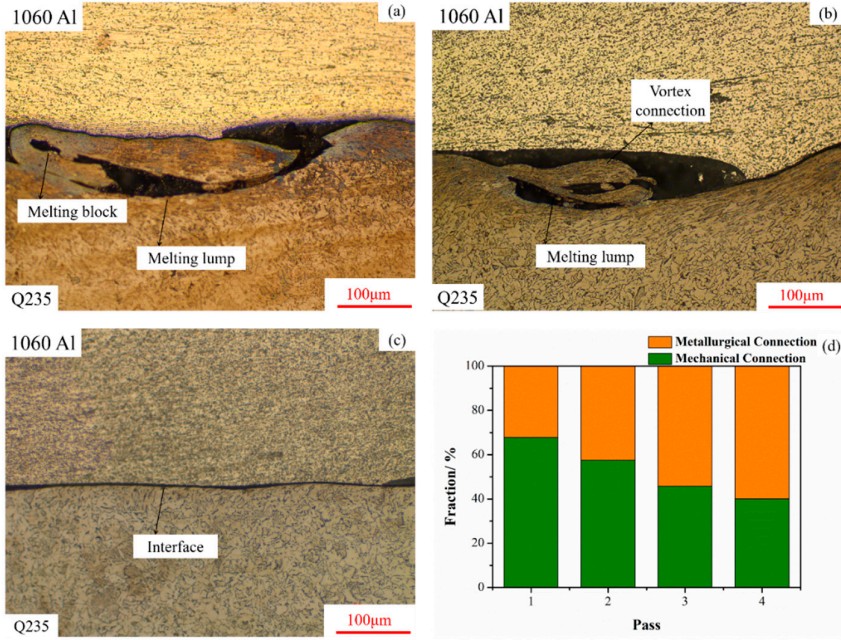

**Figure 6.** Interface connection mechanism of composite plate: (**a**) hook connections; (**b**) vortex connections; (**c**) metallurgical bondings; (**d**) Fractions for mechanical connections and metallurgical bondings to the interface of composite plate after FSP with different passes.

Figure 7 shows the interface of composite plate after FSP with different passes. With the increase of passes, the thickness of IMCs layers grows. The interface is flat and straight with just single-pass FSP (Figure 7a). In other words, FSP cannot repair the interface of composite plate with the single-pass FSP effectively. Compared with single-pass FSP, there is a little more thickness of IMCs in the interface with two passes. However, the effect of repair with two passes is still insufficient in contrast to the whole interface. That is to say, the effect of repair is not obvious (Figure 7b). The thickness of IMCs clearly increases with three passes. There is a superior effect of repair with the discontinuous mechanical connections (Figure 7c). The interface is so thick that the bonding strength of the interface is low. Too many passes make the heat input too large, which will make the interface too thick. Due to the existence of too thick interface, the original discontinuous mechanical connections have been connected together. The properties of composite plate must be affected with the too thick interface and continuous connections (Figure 7d). It is clear that too many passes will cause the deterioration of properties for the interface.

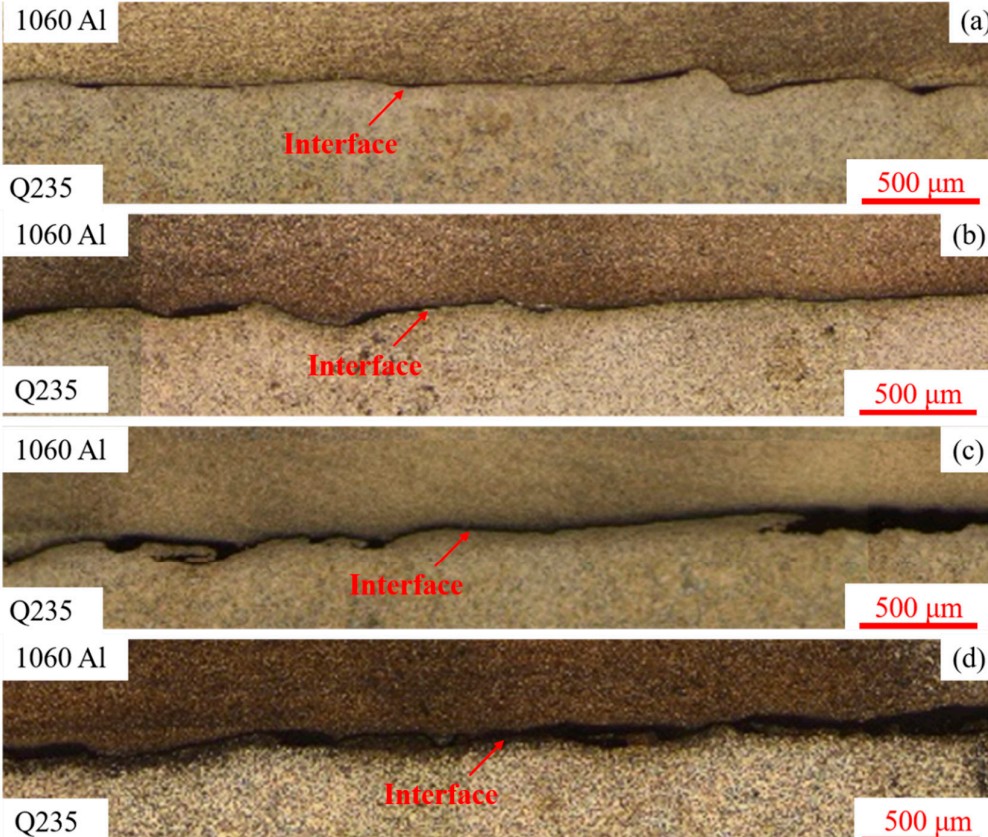

**Figure 7.** Interfaces of composite plate after FSP with different passes: (**a**) single pass; (**b**) two passes; (**c**) three passes; (**d**) four passes.

In summary, FSP can reduce or even eliminate steel cuttings, holes and cracks caused by explosive welding through the plastic flow of aluminum. However, the repair of single-pass FSP is nothing like enough. The M-FSP can effectively increase the heat input and make the IMCs layer grow and thicken effectively. Meanwhile, too many passes also deteriorate the performances of composite plate.

### 3.3. Mechanical Properties

Figure 8 shows the fracture morphologies and mechanical property curves for the shear specimens and the shear strengths are shown in Table 3. The fracture mode of shear specimens exhibits typical brittle fracture (Figure 8a). The curve of single-pass shows the phenomenon of yielding caused by the tunnel defects under the load. While the curve of two passes and three passes exhibit the brittle fracture more obviously (Figure 8b). Compared with the single-pass FSP, M-FSP has a uniform refinement effect on the aluminum and finally improves the strength of aluminum. When subjected to load, cracks preferentially appear at the interface and the samples break due to the fracture of the IMCs layer. The shear strengths of composite plate after FSP with different passes are shown in Table 3. Due to the yield strength of 1060Al is about 35 MPa and the shear strengths of specimens range from 28.87 MPa to 33.37 MPa, which show that the shear properties of composite plate after FSP are in line with requirements. From the shear strength and curves of different passes, it can be seen that when the passes are three and four, the shear strength is higher than those are one and two. Although the strength increase is not obvious, the shear properties of composite plate are still improved by M-FSP.

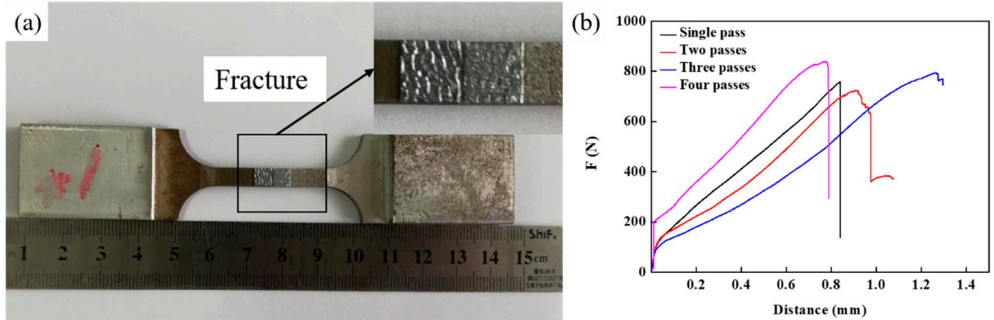

**Figure 8.** Fracture morphologies and strength of shear specimens: (**a**) fracture morphologies of shear specimens; (**b**) mechanical property curves of shear specimens.

**Table 3.** Shear strength of samples after FSP with different passes.

| Pass | S (mm²) | Force (N) | τ (MPa) |
|------|---------|-----------|---------|
| 1 | 25.10 | 793.6 | 31.62 |
| 2 | 25.05 | 723.4 | 28.87 |
| 3 | 24.90 | 829.5 | 33.31 |
| 4 | 25.13 | 838.6 | 33.37 |

The SEM and EDS analysis of shear specimens are shown in Figure 9. Strippings are found in the fracture surfaces, as shown in Figure 9a–c. Analysis shows that these strippings are formed by the failure of the original mechanical connections at the interface under the load. The second phase particles are also obviously seen in the SEM. The EDS result shows that Si element is contained in these particles, which indicates that the second phase particles are Al-Si compounds (Figure 9d). A small amount of dispersed Al-Si compounds can improve mechanical properties of composite plate.

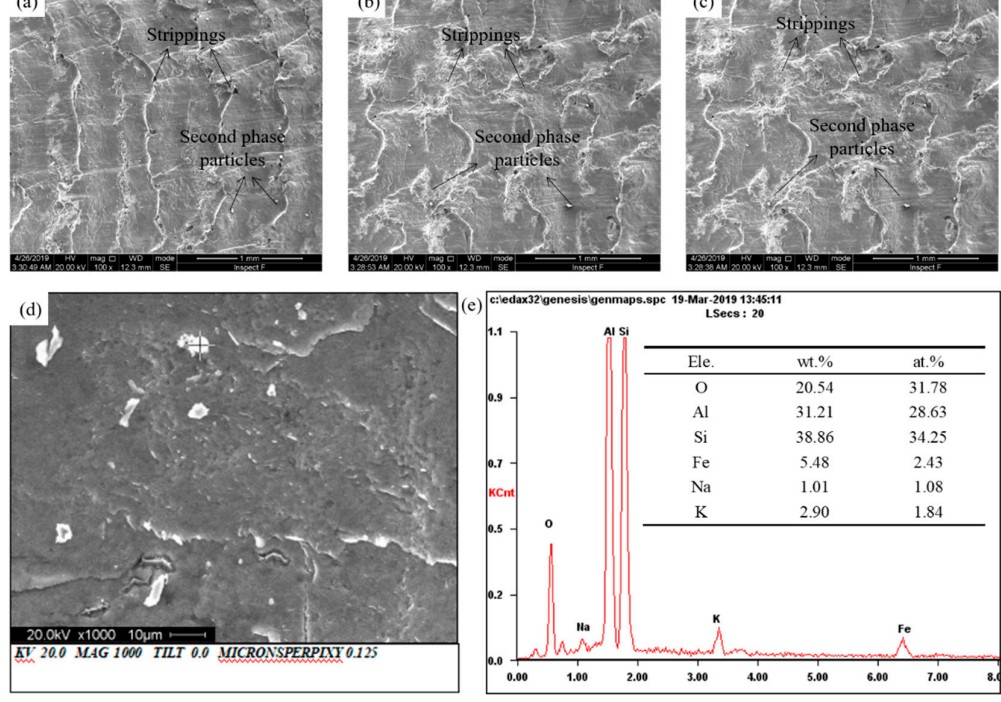

**Figure 9.** Scanning electron microscope (SEM) analysis of shear specimens: (**a**) single pass; (**b**) two passes; (**c**) three passes; (**d**) and (**e**) SEM and Energy Dispersive Spectroscopy (EDS) analysis of second phase particles.

The shear mechanical properties of BM are also carried out, as shown in Figure 10. The results show that the phenomenon of yield occurs on the aluminum side of BM. The maximum failure tensile force is 1777 N. However, the failure strength is not the shear strength, while is the tensile strength of 1060Al. The failure strength is 118.47 MPa, which is conformed to 1060Al standard specification. The fracture position of BM without FSP is in the aluminum side rather than at the interface. This is due to the refining effect of FSP in aluminum, which makes the aluminum obtain the grain refining strength after FSP. Therefore, FSP has an obvious repairing effect on the interface of composite plate.

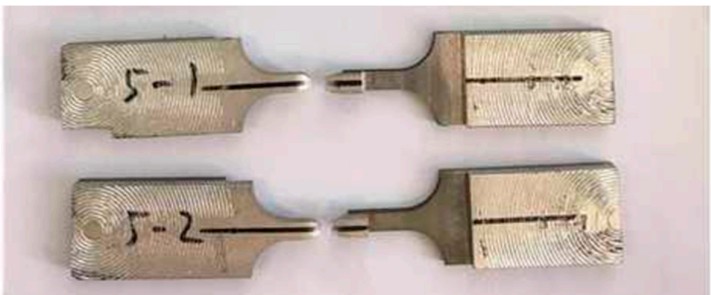

**Figure 10.** Shear specimens of BM.

The results of bending test are shown in Table 4. The typical bending specimens are shown in Figure 11. Whether repaired or not, the bending strength is superior. Mainly due to the excellent plasticity of 1060Al, there are almost no cracks during the process of bending. However, cracks can be seen in aluminum after FSP with four passes, as shown in Figure 12. The failure angle of bending is 137°. Excessive axial force of the shoulder leads to the thinning of BM with the cracking phenomenon caused by the reduction of the bearing area in bending. Therefore, it should be well controlled in the M-FSP to ensure the properties of bending.

**Table 4.** Results of bending test of composite plate.

| Pass | Crack in steel (tensile face) | Crack in aluminum (compressed face) | Crack in steel (compressed face) | Crack in aluminum (tensile face) |
|------|------------------|------------------|------------------|------------------|
| 0 | No | No | No | No |
| 1 | No | No | No | No |
| 2 | No | No | No | No |
| 3 | No | No | No | No |
| 4 | No | No | No | Yes |

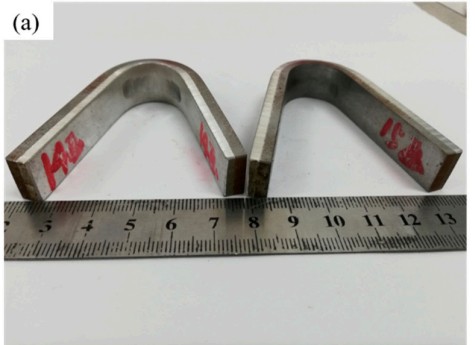
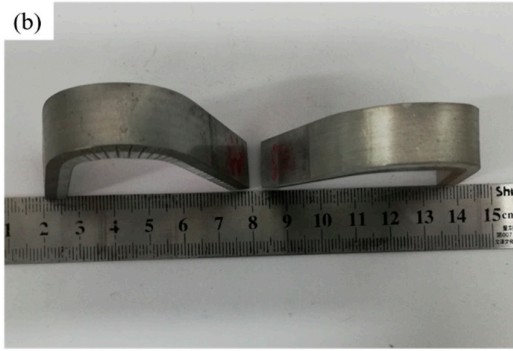

**Figure 11.** Bending specimens of composite plate after FSP with two passes: (**a**) compressed face; (**b**) tensile face

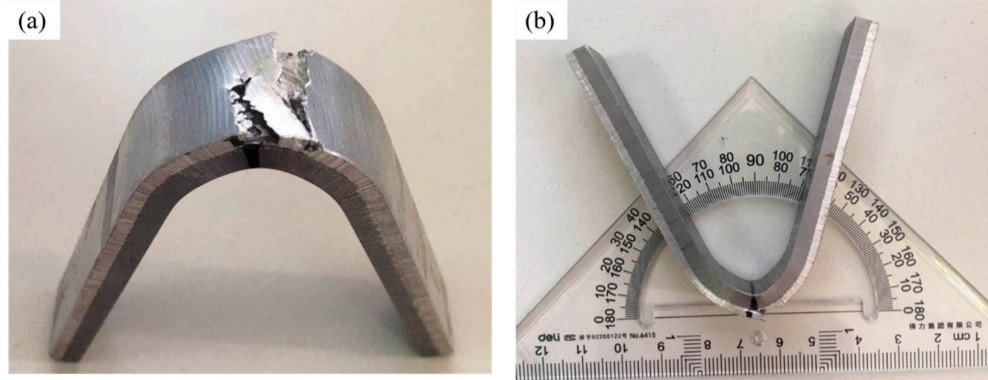

**Figure 12.** Bending specimens of composite plate after FSP with four passes: (**a**) tensile face; (**b**) bending diagram of angle measurement.

The microhardness of composite plates after FSP is shown in Figure 13. The microhardness is measured in the positive direction along the aluminum with the interface as the origin. The hardness on the interface near the aluminum side without FSP is 44.54HV$_{0.05}$. While the maximum hardness of the interface after FSP can be achieved to 53.83HV$_{0.05}$. It is believed that FSP can enhance performances of composite plate and has an obvious repairing effect. Meanwhile, FSP also has an effect of improvement on the properties of metal near the interface. The hardness of Q235 without FSP is 198.5HV$_{0.05}$ with the distance of 30μm away from the interface, while the hardness can reach to 270.9HV$_{0.05}$ with the same distance after FSP. Although just repairing in aluminum, FSP can still strengthen steel through the plastic flows of aluminum.

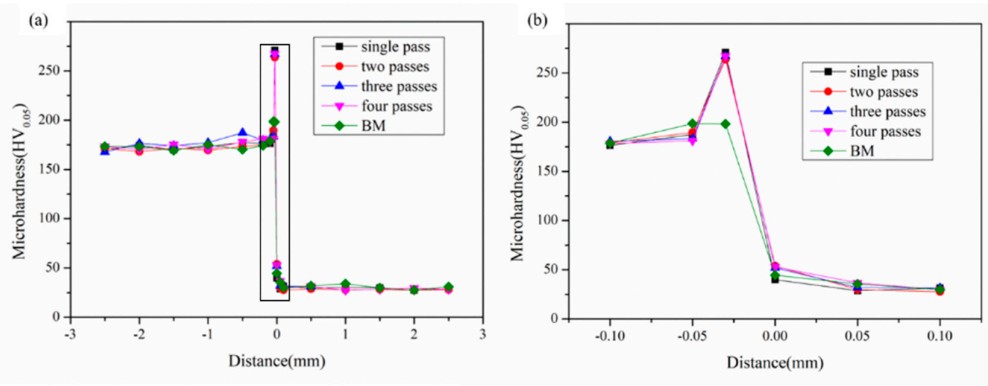

**Figure 13.** Microhardness of composite plate after FSP with different passes: (**a**) hardness of whole specimens; (**b**) hardness for the samples near the interface (partial enlargement of Figure 13a).

## 4. Discussion

The relationships between microstructures and mechanical properties for the aluminum/steel composite plate with different passes are analyzed above. The results show that M-FSP not only repair the interface but also repair the tunnel defects in the aluminum side. However, too many passes also have negative effects on the interface and metal stirred, such as metal thinning and toe flashes. Therefore, the fracture mechanism of interface for the 1060Al/Q235 composite plate is obtained by studying the relationship between structures and properties of aluminum/steel composite plate after FSP with different passes. The failure mechanism can be divided into the following three categories:

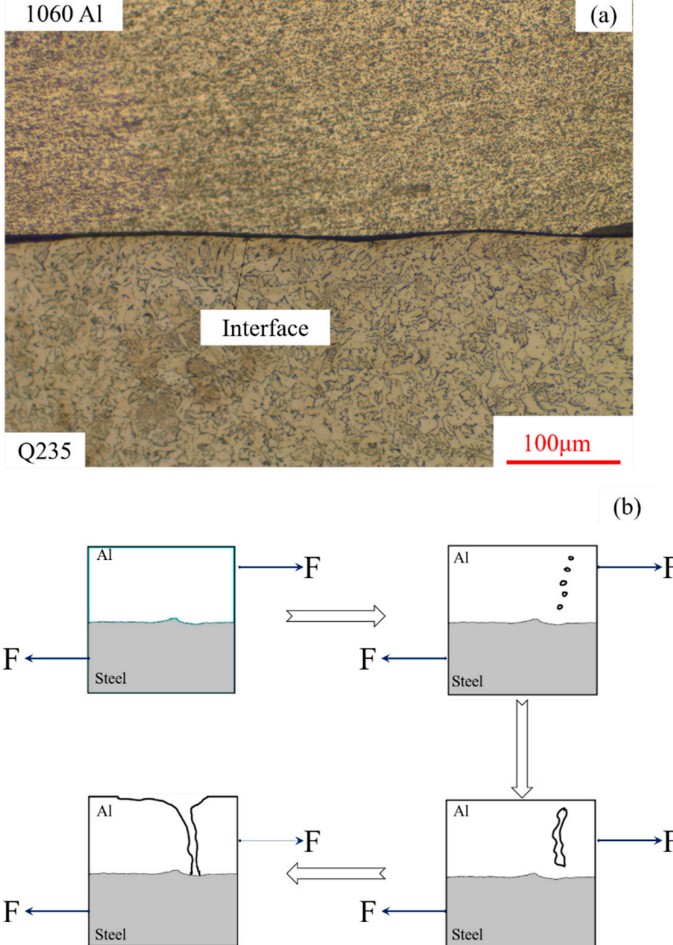

**Figure 14.** Failure mechanism of thin and flat interface: (**a**) physical diagram; (**b**) failure diagram of flat interface.

Firstly, when the interface of aluminum/steel composite plate is thin and flat, the crack extension stress of interface is small due to the flat interface. Meanwhile, the aluminum is so soft that it is easy to occur the phenomenon of yield. Several voids will preferentially be formed in the aluminum side. Cracks are formed by the accumulation and growth of voids with the gradually increasing load. Eventually, cracks extend to the aluminum and lead to the failure of composite plate, as shown in Figure 14. This failure mostly occurs in the aluminum/steel composite plate without FSP. It is basically agreed with the yield fracture of BM in the shear test.

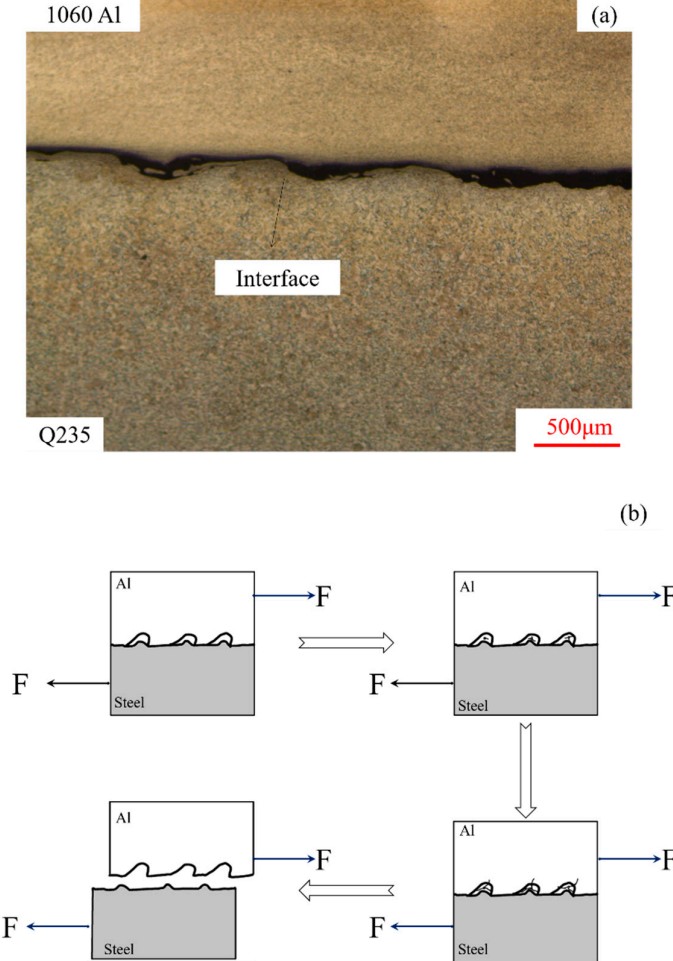

**Figure 15.** Failure mechanism of thick and discontinuous interface. (**a**) physical diagram; (**b**) failure diagram of flat interface.

Secondly, when the interface of aluminum/steel composite plate has a certain thickness and discontinuous mechanical connections, the crack extension stress of the interface is large. Several micro-cracks will preferentially be formed at the interface under the load. If the composite plate has been inadequately repaired by less passes, there could be some defects at the interface. The phenomenon of stress concentration caused by defects would accelerate the development of micro-cracks. Therefore, those micro-cracks would coalesce each other rapidly and ultimately lead to the fracture of the whole interface. However, if the composite plate has been adequately repaired by enough passes, the interface would have a certain thickness. The propagation of micro-cracks needs a path. In addition to this, those discontinuous mechanical connections improve the bonding strength of interface, which make the cracks propagate slowly. Thus, a larger load is needed to make the composite plate break. This failure occurs mostly in the aluminum/steel composite plate after repairing by FSP, which is basically consistent with the case of single, two and three passes, as shown in Figure 15.

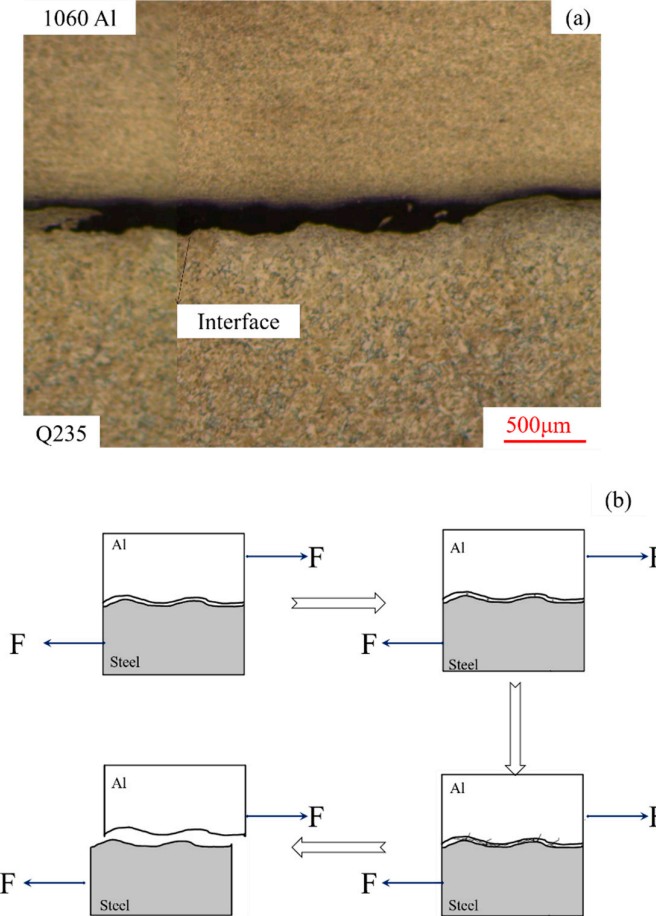

**Figure 16.** Failure mechanism of too thick and continuous interface: (**a**)physical diagram; (**b**) failure diagram of flat interface.

Thirdly, when the interface is too thick, although there are discontinuous mechanical connections at the interface, the overly thick interface connects the discontinuous mechanical connections to each other, which results in a bending and continuous interface. The crack extension stress of the interface is also large due to the bending interface. Micro-cracks will preferentially be formed at the interface under the load. Due to the thicker interface, those micro-cracks could easily bypass the mechanical connections. Therefore, the micro-cracks grow rapidly with the increasing load. Finally, the composite plate will break due to the fracture of the interface. This failure occurs more often in the composite plate after FSP with too many passes. The fracture of the aluminum/steel composite plate is basically the same as that of four passes, as shown in Figure 16.

## 5. Conclusion

In this paper, microstructures and properties for the interface of the 1060Al/Q235 composite plate are studied. By analyzing the internal relationships between microstructures and mechanical properties of the interface, the fracture mechanisms of aluminum/steel composite plate are summarized. The main conclusions are concluded:

1. Defects at the interface for the aluminum/steel composite plate could be repaired by the FSP.
2. M-FSP can increase the thickness of interface for the composite plate. Meanwhile, it can also repair the tunnel defects remained by the single-pass FSP.
3. The melting block and the melting lump in the composite plates are easy to become originals of crack. Therefore, when the interfaces of composite plate mainly consist of the metallurgical

bondings, with a certain thickness and discontinuous mechanical connections, its bonding strength is superior.

**Author Contributions:** Conceptualization, J.W. and B.L.; methodology, C.C.; validation, J.W. and B.L.; investigation, J.W. and Y.C.; data curation, Y.C.; writing—original draft preparation, J.W.; writing—review and editing, C.C.; supervision, J.W.; project administration. All authors have read and agreed to the published version of the manuscript.

**Funding:** This work was funded by the National Natural Science Foundation of China (No. 51505293), the Natural Science Foundation of Jiangsu Province (No. BK20190684), the Natural Science Research of the Jiangsu Higher Education Institutions of China (No. 18KJB460016) and the Introduce Talent Special Funding for Scientific Research at Nanjing Tech University (No. 39802124).

**Conflicts of Interest:** The authors declare no conflict of interest.

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
