# Peer review of "Effects of Multi-Pass Friction Stir Processing on Microstructures and Mechanical Properties of the 1060Al/Q235 Composite Plate"

_metals, doi:10.3390/met10030298_

Round 1

Reviewer 1 Report

The effect of multi-pass friction stir processing on the microstructural and the mechanical properties of 1060Al/Q235 explosive composite plates was studied in this research. Solid-state processing and welding are very interesting research fields and many works have been published in these areas.

An experimental work supported by a large range of characterisation techniques was conducted by the authors. However, the reviewer thinks that the effectiveness of the proposed FSP procedure is very questionable. Al-Steel explosive composite plates were processed by FSP and their microstructure and mechanical properties were characterised. When the authors assessed the mechanical properties of the welded composites, their properties were much better than those of the processed samples. In fact, while the non-processed samples were found to fail outside the joining region of the specimens, specifically in the Al side, the processed samples (regardless of the number of processing passes) fractured in the joining region, for a much lower testing load. So, the reviewer does not understand the interest of processing the welded samples, since the properties of the interface were severely affected, resulting in a very premature failure of the specimens (comparing to the non-processed specimens).

According to the reviewer, the topic addressed in the previous paragraph is the main issue that may affect the publication of this work. If the authors are able to properly explain the interest of this procedure, and consequently, the relevance of this research, they should also consider the following comments:

In the Experimental Materials and Methods, the authors refer that the processing parameters were defined according to the previous study. However, this study is not referenced.

The friction stir processing tool must be characterised in the Experimental Materials and Methods: Shoulder geometry and dimension; Pin geometry and dimension (diameter and length).

The quality of the macrographs presented in Fig. 2 must be improved in order to enable to better observe the entire macrostructure of the processed region.

The authors should compare the grain size of both welded materials at the interface of the composite, i.e. the grain size of the materials only after welded vs the grain size of the materials after welded and friction stir processed.

The authors refer:

“Such a large steel cutting has a severely disadvantage effect on the strength of the interface.”

How do the authors support this statement?

The authors report the formation of oxides at the interface of the composite during explosive welding. How do the authors know that these oxides were formed during welding and not after welding, i.e. between the preparation of the samples and the SEM analysis?

The captions of some of the figures must be improved in order to better explain what is really presented there (for example, Fig. 3 and Fig. 4).

The authors refer:

“Actually, metallurgical bondings are made up of IMCs layers. However, these IMCs separated from the interface are disadvantageous to the properties of composite plate.”

What do the authors mean with “IMCs separated from the interface”?

The graph presented in Fig. 5d must be better explained, specifically, the way it was achieved.

The authors refer:

“The thickness of IMCs clearly increases with three passes. There is a superior effect of repair with the discontinuous mechanical connections (Fig.6(c)).”

What do the authors mean with “discontinuous mechanical connections”?

The quality of the macrographs presented in Fig. 6 must be improved. Some arrows with text indicating what the authors are referring to would help to better understand the macrographs and their explanation.

The authors refer:

“In summary, FSP can reduce or even eliminate the defects caused by explosive welding through the plastic flow of aluminum.”

What kind of defects are the authors referring to?

The authors refer:

“The microhardness is measured in the positive direction along the aluminum with the interface as the origin. The hardness of IMCs on the interface without FSP is 44.54HV0.05. While the maximum hardness of the interface after FSP can be achieved to 53.83HV0.05.”

Considering the typical hardness of Al-Fe intermetallic phases, how can the authors refer “hardness of IMCs on the interface without FSP is 44.54HV0.05”?

One of the conclusions of the authors is:

“1. FSP can repair defects of the interface for aluminum/steel composite plate and improve the bonding strength of composite plate.”

All the processed samples failed at the bonding interface, while the non-processed samples failed outside this region. So, the reviewer does not understand in which way the FSP improves the mechanical properties of the welded samples.

Author Response

Response to the Reviewer 1# Comment

Question 1: An experimental work supported by a large range of characterisation techniques was conducted by the authors. However, the reviewer thinks that the effectiveness of the proposed FSP procedure is very questionable. Al-Steel explosive composite plates were processed by FSP and their microstructure and mechanical properties were characterised. When the authors assessed the mechanical properties of the welded composites, their properties were much better than those of the processed samples. In fact, while the non-processed samples were found to fail outside the joining region of the specimens, specifically in the Al side, the processed samples (regardless of the number of processing passes) fractured in the joining region, for a much lower testing load. So, the reviewer does not understand the interest of processing the welded samples, since the properties of the interface were severely affected, resulting in a very premature failure of the specimens (comparing to the non-processed specimens).

Response 1: Thank you for the reviewer’s useful comment. The shear strength for the sample without FSP (BM) does not obtained owing to the sample failure in the aluminum side, while not in the interface. However, when the samples experience the friction stir processing, the sample failure at the interface. Such two samples (samples without M-FSP and with M-FSP) have the same base metal, but with different condition. The grain size of base metal for the aluminum side without M-FSP is about 22.6μm, while the grain size of base metal for the aluminum side after M-FSP is about 3.6μm. That is due to the refining effect of FSP in aluminum, which makes the aluminum get grain refining strength after FSP. M-FSP has a uniform refinement effect on the aluminum and finally improves the strength of aluminum. Therefore, the strength of base metal for the aluminum side after M-FSP is much higher than that before M-FSP. So, when testing the shear strength for the samples after M-FSP, the samples failure at the interface. That is not M-FSP does not improve the shear strength, but only because the refinement strengthening for the base metal in the aluminum side.

Question 2: In the Experimental Materials and Methods, the authors refer that the processing parameters were defined according to the previous study. However, this study is not referenced. The friction stir processing tool must be characterised in the Experimental Materials and Methods: Shoulder geometry and dimension; Pin geometry and dimension (diameter and length).

Response 2: Thank you for the reviewer’s useful comment. We have added the relevant sentences in the Experimental materials and methods section. The added sentences are shown as below:

“The axial force during M-FSP process is 10kN. The FSW tool was made of W-Re alloy, and the FSW tool geometry consisted of a threaded conical pin, with three flats (7mm diameter) and a spiral (scroll) shoulder with a diameter of 25mm. The tilt angle for the FSW tool during welding is 2°. The pin tool plunge depth is 2.5mm.”

Question 3: The authors should compare the grain size of both welded materials at the interface of the composite, i.e. the grain size of the materials only after welded vs the grain size of the materials after welded and friction stir processed.

Response 3: Thank you for the reviewer’s useful comment. We have added the grain size of the materials only after welded vs the grain size of the materials after welded and friction stir processed. The added sentence is shown as below:

“The grain size of 1060Al after M-FSP is about 3.6μm, much smaller than that before M-FSP (22.6μm).”

Question 4: The authors refer:

“Such a large steel cutting has a severely disadvantage effect on the strength of the interface.”

How do the authors support this statement?

Response 4: Thank you for the reviewer’s useful comment. In the introduction section, we wrote that defects such as steel cuttings and microcracks caused by explosion weaken the strength of composite. Normally, the average grain size for the materials after friction stir processed is about 10μm. The steel cuttings with the size of 45μm is much larger than that of the average grain size. Therefore, such a large steel cutting has a severely disadvantage effect on the strength of the interface.

Question 5: The authors report the formation of oxides at the interface of the composite during explosive welding. How do the authors know that these oxides were formed during welding and not after welding, i.e. between the preparation of the samples and the SEM analysis?

Response 5: Thank you for the reviewer’s useful comment. During the explosive welding process, the two metal plates are joined together in the atmosphere environment, not in the vacuum environment. Beside this, the surfaces of explosive composite plates still have some oxide scales owing to the massive production of industrial manufacture. Last but not least, such oxides are not of the case study and can be found easily in the interface. Therefore, these oxides at the interface of the composite were formed during explosive welding.

Question 6: The captions of some of the figures must be improved in order to better explain what is really presented there (for example, Fig. 3 and Fig. 4).

Response 6: Thank you for the reviewer’s useful comment. We have rewrite the captions of Fig.3 and Fig.4, and the revised captions are shown as below:

“Fig.3. Microstructures of composite plate after FSP with different passes: (a) repair zone; (b) BM of aluminum; (c) BM of steel; (d) SZ; (e) TMAZ in the aluminum side; (f) HAZ in the steel side and (g) TEM image near the interface”

“Fig.4. Comparisons between the unrepaired interface and the repaired interface: (a) and (b) unrepaired interface; (c) repaired interface; (d) and (e) SEM and EDS analysis of the unrepaired interface; (f) and (g) SEM and EDS analysis of the repaired interface”

Question 7: The authors refer:

“Actually, metallurgical bondings are made up of IMCs layers. However, these IMCs separated from the interface are disadvantageous to the properties of composite plate.”

What do the authors mean with “IMCs separated from the interface”?

Response 7: Thank you for the reviewer’s useful comment. The aluminum/steel composite plate interface consists of mechanical connections and metallurgical bondings formed by IMC. However, due to the high energy and unstable characteristics of explosive welding, parts of the IMC in the interface metallurgical bonding layer separated and embedded in the softer aluminum, forming melting lumps and melting blocks, as shown in Figure 5 (a) and Figure 5 (b). These separated IMCs are weak points of mechanical properties, and cracks often initiate at such places. Therefore, these IMCs separated from the interface are detrimental to the performance of the composite board.

Question 8: The authors refer:

“The thickness of IMCs clearly increases with three passes. There is a superior effect of repair with the discontinuous mechanical connections (Fig.6(c)).”

What do the authors mean with “discontinuous mechanical connections”?

Response 8: Thank you for the reviewer’s useful comment. As can be seen from Fig.15, When the interface of aluminum/steel composite plate has a certain thickness and discontinuous mechanical connections, the crack extension stress of the interface is large. Those discontinuous mechanical connections improve the bonding strength of the interface, which make the cracks propagate slowly. Thus, a larger load is needed to make the composite plate break.

Question 9: The quality of the macrographs presented in Fig. 6 must be improved. Some arrows with text indicating what the authors are referring to would help to better understand the macrographs and their explanation.

Response 9: Thank you for the reviewer’s useful comment. We have added the arrows to indicate the location of interface more clearly so that it is much more convenience for us to understand the thickness of interface with different passes. The revised figure is shown as below:

Fig.6. The interfaces of composite plate after FSP with different passes: (a) single pass; (b) two passes; (c) three passes; (d) four passes

Question 10: The authors refer:

“In summary, FSP can reduce or even eliminate the defects caused by explosive welding through the plastic flow of aluminum.”

What kind of defects are the authors referring to?

Response 10: Thank you for the reviewer’s useful comment. These defects are steel cuttings, holes and cracks at the interfaces owing to the explosive welding process, as written in the Introduction part. In order to eliminate the mistakes, we have revised sentences as below:

“In summary, FSP can reduce or even eliminate steel cuttings, holes and cracks caused by explosive welding through the plastic flow of aluminum.”

Question 11: The authors refer:

“The microhardness is measured in the positive direction along the aluminum with the interface as the origin. The hardness of IMCs on the interface without FSP is 44.54HV0.05. While the maximum hardness of the interface after FSP can be achieved to 53.83HV0.05.”

Considering the typical hardness of Al-Fe intermetallic phases, how can the authors refer “hardness of IMCs on the interface without FSP is 44.54HV0.05”?

Response 11: Thank you for the reviewer’s useful comment. As can be seen from Fig.13, the hardness with the value of 44.54HV0.05 is near the aluminum side. The maximum hardness of IMC can reach to 270.9HV0.05 after M-FSP. Therefore, in order to remove the mistake, we have revised the sentence. The revised sentence is shown as below:

“The hardness on the interface near the aluminum side without FSP is 44.54HV0.05. While the maximum hardness of the interface after FSP can be achieved to 53.83HV0.05.”

Question 12: One of the conclusions of the authors is:

“1. FSP can repair defects of the interface for aluminum/steel composite plate and improve the bonding strength of composite plate.”

All the processed samples failed at the bonding interface, while the non-processed samples failed outside this region. So, the reviewer does not understand in which way the FSP improves the mechanical properties of the welded samples.

Response 12: Thank you for the reviewer’s useful comment. The shear strength for the sample without FSP (BM) does not obtained owing to the sample failure in the aluminum side, while not in the interface. However, when the samples experience the friction stir processing, the sample failure at the interface. Such two samples (samples without M-FSP and with M-FSP) have the same base metal, but with different condition. The grain size of base metal for the aluminum side without M-FSP is about 22.6μm, while the grain size of base metal for the aluminum side after M-FSP is about 3.6μm. That is due to the refining effect of FSP in aluminum, which makes the aluminum get grain refining strength after FSP. Therefore, the strength of base metal for the aluminum side after M-FSP is much higher than that before M-FSP. So, when testing the shear strength for the samples after M-FSP, the samples failure at the interface. That is not M-FSP does not improve the shear strength, but only because the refinement strengthening for the base metal in the aluminum side. In order to eliminate the mistake, we have revised the conclusion and revised conclusion is shown as below:

“1. FSP can repair defects of the interface for the aluminum/steel composite plate.”

Reviewer 2 Report

It is possible that FSP can repair defects of the interface for aluminum/steel composite plate. Maybe the process of destruction of joints for various M-FSP passes  is as in the drawings (Fig 13, 14, 15). However, by presenting the results of the research on the graph (Fig 7 b) and in the table (Table 3), the authors strongly confused the understanding of the text.

The course of the curves on the graph (Fig 7) shows that the sample with one pass has the greatest shear force, the sample with three passes has the average force, the sample with two passes has the lowest force. There is no graph for the sample with four passes. Destruction in any case occurs at a force less than 800 N !.

In Table 3, on the other hand, the sample with three passes (829.5 N) has more strength than with one pass (793.6 N). This is not consistent with the Fig 7 chart data!(?)

Therefore, the sentence on p. 11 is not true: “From the shear strength and curves of different passes, it can be seen that when the passes are three and four, the shear strength is higher than those are one and two”.

In addition, test results for the sample without FSP (BM) are given. The destruction force of such a sample was 1777 N. That's right: “The failure strength is not the shear strength, while is the yield strength of 1060Al”.

From what above rather follows that FSP reduced the bonding strength of composite plate and did not improve the strength.

In this context, it is difficult to understand the second part of the first conclusion: "FSP can ... improve the bonding strength of composite plate". It does not follow from the cited test results!

I suggest checking the curves in Fig. 7, sorting out the measurement data (Table 3) and formulating the conclusions again.

In addition, the number of samples provided for shear testing was not given in the paper. One for each case?

Author Response

Response to the Reviewer 2# Comment

Question 1: The course of the curves on the graph (Fig 7) shows that the sample with one pass has the greatest shear force, the sample with three passes has the average force, the sample with two passes has the lowest force. There is no graph for the sample with four passes. Destruction in any case occurs at a force less than 800 N !.

In Table 3, on the other hand, the sample with three passes (829.5 N) has more strength than with one pass (793.6 N). This is not consistent with the Fig 7 chart data!(?)

Therefore, the sentence on p. 11 is not true: “From the shear strength and curves of different passes, it can be seen that when the passes are three and four, the shear strength is higher than those are one and two”.

In addition, test results for the sample without FSP (BM) are given. The destruction force of such a sample was 1777 N. That's right: “The failure strength is not the shear strength, while is the yield strength of 1060Al”.

From what above rather follows that FSP reduced the bonding strength of composite plate and did not improve the strength.

In this context, it is difficult to understand the second part of the first conclusion: "FSP can ... improve the bonding strength of composite plate". It does not follow from the cited test results!

I suggest checking the curves in Fig. 7, sorting out the measurement data (Table 3) and formulating the conclusions again.

Response 1: Thank you for the reviewer’s useful comment. According to the non-standard shear samples, we can not obtain the stress-strain curve, but only get the load-displacement curve from the machine. We have checked the mechanical property curves of shear specimens and added the curves of shear specimens after four passes. The shear strength of samples after FSP with different passes are also corrected in consistent with the datum in Fig.8(b). Results show that when the passes are three and four, the shear strength is higher than those are one and two. The revised figure and the revised table are shown as below:

Fig.8. Fracture morphologies and strength of shear specimens: (a) fracture morphologies of shear specimens; (b) mechanical property curves of shear specimens

Table 3. Shear strength of samples after FSP with different passes

Pass

S (mm2)

Force (N)

τ (MPa)

1

25.10

793.6

31.62

2

25.05

723.4

28.87

3

24.90

829.5

33.31

4

25.13

838.6

33.37

The shear strength for the sample without FSP (BM) does not obtained owing to the sample failure in the aluminum side, while not in the interface. However, when the samples experience the friction stir processing, the sample failure at the interface. Such two samples (samples without M-FSP and with M-FSP) have the same base metal, but with different condition. The grain size of base metal for the aluminum side without M-FSP is about 22.6μm, while the grain size of base metal for the aluminum side after M-FSP is about 3.6μm. That is due to the refining effect of FSP in aluminum, which makes the aluminum get grain refining strength after FSP. Therefore, the strength of base metal for the aluminum side after M-FSP is much higher than that before M-FSP. So, when testing the shear strength for the samples after M-FSP, the samples failure at the interface. That is not M-FSP does not improve the shear strength, but only because the refinement strengthening for the base metal in the aluminum side. In order to eliminate the mistake, we have revised the conclusion and revised conclusion is shown as below:

“1. FSP can repair defects of the interface for the aluminum/steel composite plate.”

Question 2: In addition, the number of samples provided for shear testing was not given in the paper. One for each case?

Response 2: Thank you for the reviewer’s useful comment. We have added some sentences in the Experimental materials and methods section to give an explanation for this. The added sentences are shown as below:

“Three shear testing samples and two bending testing samples were tested for each group in order to give a better reliability for the testing datum.”

Reviewer 3 Report

The subject matter of the article is interesting and current and is the main asset of the work. The work is valuable to scientists dealing with friction stir processing.

Detailed comments:

The term "explosive composite plate" is misleading. Explosive welding was used to make the composite plate, but the plate itself does not have explosive properties. I recommend removing the word “explosive” throughout the article, including the title, and only in the presentation of the material should it be explained that the bonding was obtained by the explosive welding method 

In the Introduction, the authors wrote: "The microstructures are observed by optical microscopy, scanning electron microscopy and transmission electron microscopy. The shear fracture morphologies are researched to analyze the fracture mechanism. The bending tests are carried out to investigate bending properties of the composite plate after FSP with different passes. The hardness distributions of the interface in the repair zone are measured by microhardness test. The experiments control a single variable, which can study the influence of different passes for FSP"

This text is completely unnecessary, because it only characterizes the type of instrumentation that can be used to examine changes resulting from FS Processing. There is nothing insightful and significant in the above description in the context of the purpose of the work.

“The overlap rate is 100%”. Should be „was” The authors do not provide data on the shape and dimensions of the tip and tool shoulder used in friction stir processing. What material was the tool made of? The authors wrote: “The black particles are the oxides of aluminum and steel by melting and reacting with oxygen in the air during explosive welding”. – The term “oxides of steel” is incorrect, should be “oxides of iron” The authors state in paragraph 3.3 that "when the passes are three and four, the shear strength is higher than those are one and two". However, they leave this fact without comment. Is it due to the presence of tunnel defects or e.g. the presence of brittle oxides? In paragraph 3.2. the authors showed that "too many passes also have a negative effect on the properties of composite plate," and then in paragraph 3.3. they stated that "when the passes are three and four, the shear strength is higher than those are one and two". There is a contradiction here. It is necessary to supplement the text with a detailed explanatory comment. Table No. 4 is completely unnecessary due to identical results in all variants. I recommend removing Table No. 4. Instead of Table No. 4, just one comment sentence is necessary, that no cracks were found in individual tests

Author Response

Response to the Reviewer 3# Comment

Question 1: The term "explosive composite plate" is misleading. Explosive welding was used to make the composite plate, but the plate itself does not have explosive properties. I recommend removing the word “explosive” throughout the article, including the title, and only in the presentation of the material should it be explained that the bonding was obtained by the explosive welding method.

Response 1: Thank you for the reviewer’s useful comment. We have removed the word “explosive” throughout the article. Now it is the “the 1060Al/Q235 composite plate”.

Question 2: In the Introduction, the authors wrote: "The microstructures are observed by optical microscopy, scanning electron microscopy and transmission electron microscopy. The shear fracture morphologies are researched to analyze the fracture mechanism. The bending tests are carried out to investigate bending properties of the composite plate after FSP with different passes. The hardness distributions of the interface in the repair zone are measured by microhardness test. The experiments control a single variable, which can study the influence of different passes for FSP". This text is completely unnecessary, because it only characterizes the type of instrumentation that can be used to examine changes resulting from FS Processing. There is nothing insightful and significant in the above description in the context of the purpose of the work.

Response 2: Thank you for the reviewer’s useful comment. We have deleted the relevant sentences according to the reviewer’s comment.

Question 3: “The overlap rate is 100%”. Should be “was”.

Response 3: Thank you for the reviewer’s comment. We have changed the word in the sentence and the revised sentence is shown as:

“The overlap rate was 100%.”

Question 4: The authors do not provide data on the shape and dimensions of the tip and tool shoulder used in friction stir processing. What material was the tool made of?

Response 4: Thank you for the reviewer’s comment. We have added the relevant sentences in the Experimental materials and methods section. The added sentences are shown as below:

“The axial force during M-FSP process is 10kN. The FSW tool was made of W-Re alloy, and the FSW tool geometry consisted of a threaded conical pin, with three flats (7mm diameter) and a spiral (scroll) shoulder with a diameter of 25mm. The tilt angle for the FSW tool during welding is 2°. The pin tool plunge depth is 2.5mm.”

Question 5: The authors wrote: “The black particles are the oxides of aluminum and steel by melting and reacting with oxygen in the air during explosive welding”. – The term “oxides of steel” is incorrect, should be “oxides of iron”.

Response 5: Thank you for the reviewer’s comment. We have corrected the mistake. The revised sentence is shown as below:

“The black particles are the oxides of aluminum and iron by melting and reacting with oxygen in the air during explosive welding.”

Question 6: The authors state in paragraph 3.3 that "when the passes are three and four, the shear strength is higher than those are one and two". However, they leave this fact without comment. Is it due to the presence of tunnel defects or e.g. the presence of brittle oxides? In paragraph 3.2. the authors showed that "too many passes also have a negative effect on the properties of composite plate," and then in paragraph 3.3. they stated that "when the passes are three and four, the shear strength is higher than those are one and two". There is a contradiction here. It is necessary to supplement the text with a detailed explanatory comment.

Response 6: Thank you for the reviewer’s comment. M-FSP, especially when the passes are three and four, can repair the tunnel defects remained by the single-pass FSP. Beside this, the quantities and area of holes in the repair zone for 1060Al/Q235 composite plate after M-FSP decrease remarkably when the passes are three. Therefore, when the passes are three and four, the shear strength is higher than those are one and two due to the eliminating of tunnel defects and holes.

The mechanical properties, especially for the shear strengths of the composite plates depend largely on the IMCs layers, especially for the thickness and the morphology of the IMCs layer. With the increase of FSP passes, the accumulated heat input increases and finally the thickness of IMCs increases. Therefore, too many passes also have a negative effect on the properties of composite plate

Question 7: Table No. 4 is completely unnecessary due to identical results in all variants. I recommend removing Table No. 4. Instead of Table No. 4, just one comment sentence is necessary, that no cracks were found in individual tests.

Response 7: Thank you for the reviewer’s comment. However, one of the bending samples failure during the bending test, as listed in Table 4. Also, in the test, we have written that cracks can be seen in aluminum after FSP with four passes. Therefore, I think it is still necessary to retain the Table 4 so that it is convenient for the readers to find the failed samples easily.

Reviewer 4 Report

Dear Authors,

your study showed the microstructural and mechanical aspects of multi-pass friction stir processing of 1060Sl/Q235 explosive composite plate, which could be an interesting point for lightweight structures. However, the presented results are not enough for the scientific novel publication.

The following points can maybe improve the quality of the work:

In the experimental part: Lack of information about the FSP´s tool (shoulder, pin, …) Lack of information about Description of the FSP´s load Why did you decide for 4 Passes in the M-FSP process? Could it show different results after 5th pass? Description of Fig. 1 is not correct.

Results and Discussion part: In Figure 3, the TMAZ area of the Q235 is not shown, please add this part with a related explanation. In Figure 4; please show the point of EDX-measurement? Please modify the manuscript and prevent using the words like” to some extent”, in order to show the quantity. In Figure 12, Please show the measurement points in the real FSPed samples. In the discussion part: line4, “However, too many passes also have negative effects on …” ; What do you mean by too many passes? Did you try to apply 5 or 6 passes?

Author Response

Response to the Reviewer 4# Comment

Question 1: In the experimental part: Lack of information about the FSP´s tool (shoulder, pin, …) Lack of information about Description of the FSP´s load

Response 1: Thank you for the reviewer’s useful comment. We have added the relevant information about the FSP’s load and tool geometry. The added sentences are shown as below:

“The axial force during M-FSP process is 10kN. The FSW tool was made of W-Re alloy, and the FSW tool geometry consisted of a threaded conical pin, with three flats (7mm diameter) and a spiral (scroll) shoulder with a diameter of 25mm. The tilt angle for the FSW tool during welding is 2°. The pin tool plunge depth is 2.5mm.”

Question 2: Why did you decide for 4 Passes in the M-FSP process? Could it show different results after 5th pass? Description of Fig. 1 is not correct.

Response 2: Thank you for the reviewer’s useful comment. We only try the four passes of M-FSP according to the obvious toe flashes. Beside this, the thickness of the aluminum plate becomes thinner gradually. Last but not least, Bozzi et al. found that when the thickness of IMCs is 8μm, mechanical properties of the composite plate are the best; While the thickness increases to 42μm, mechanical properties of the composite plate are the worst. Therefore, when the passes are three and four in this study, the shear strength obtain the highest, while with the increase of passes, for example, five and six passes, they have a negative effect on the properties of composite plate. Therefore, only four passes friction stir processing have been applied in this study.

According to the description of Fig. 1 was not correct, we have refigured the figure and revised the caption. The added sentences and the figure are shown as below:

Fig.1. Dimensions of mechanical samples:

(a) bending specimen and (b) shear specimen (unit: mm)

Fig.2 shows the 1060Al/Q235 composite plate after M-FSP with different passes. The repair zone for the 1060Al/Q235 composite plate after M-FSP with different passes are well-formed with no defects in it. However, with the increase of pass, especially for the passes are four, there shows obvious toe flashes after friction stir processing, as shown in Fig.2(e) with the red arrows. Beside this, according to the continuous axial force resulting in the toe flashes after M-FSP, the thickness of the aluminum plate becomes thinner gradually. Therefore, only four passes friction stir processing have been applied in this study.

Fig.2 The 1060Al/Q235 composite plate after M-FSP with different passes: (a) the 1060Al/Q235 composite plate after M-FSP; (a) single pass; (b) two passes; (c) three passes; (d) four passes”

Question 3: Results and Discussion part: In Figure 3, the TMAZ area of the Q235 is not shown, please add this part with a related explanation.

Response 3: Thank you for the reviewer’s useful comment. We have refigured the figure 3 and added the whole repair zone in it. Beside this, we use the red frame to mark the enlarged area of the following typical zone. However, owing to the pin just inserts in the aluminum side, while not inserts in the steel side. Therefore, only the HAZ are of the Q235 can be obtained, without the TMAZ area of the Q235, as shown in Fig.3(f). The revised figure is shown as below:

Fig.3. Microstructures of composite plate after FSP with different passes: (a) repair zone; (b) BM of aluminum; (c) BM of steel; (d) SZ; (e) TMAZ in the aluminum side; (f) HAZ in the steel side and (g) TEM image near the interface

Question 4: In Figure 4; please show the point of EDX-measurement?

Response 4: Thank you for the reviewer’s useful comment. We have used the red point to mark the point of EDX-measurement. The revised figure is shown as below:

Fig.4. Comparisons between the unrepaired interface and the repaired interface: (a) and (b) unrepaired interface; (c) repaired interface; (d) and (e) SEM and EDS analysis of the unrepaired interface; (f) and (g) SEM and EDS analysis of the repaired interface

Question 5: Please modify the manuscript and prevent using the words like” to some extent”, in order to show the quantity.

Response 5: Thank you for the reviewer’s useful comment. We have removed all the words like “to some extent” in the text.

Question 6: In Figure 12, Please show the measurement points in the real FSPed samples.

Response 6: Thank you for the reviewer’s useful comment. The microhardness is measured in the positive direction along the aluminum with the interface as the origin. That means we measure the microhardness from the aluminum side to the steel side. The microhardness is measured along the thickness of the plate. The microhardness with the distance in the negative value is in the steel side, while the microhardness with the distance in the positive value is in the aluminum side. The microhardness with the distance near the original point is in the interface, and the Fig.12(b) is the partial enlargement hardness mapping for the samples near the interface.

Question 7: In the discussion part: line4, “However, too many passes also have negative effects on …” ; What do you mean by too many passes? Did you try to apply 5 or 6 passes?

Response 7: Thank you for the reviewer’s useful comment. Too many passes can bring the plate thinning and the toe flashes for the composite plate. Beside this, Bozzi et al. found that when the thickness of IMCs is 8μm, mechanical properties of the composite plate are the best; While the thickness increases to 42μm, mechanical properties of the composite plate are the worst. Therefore, when the passes are three and four in this study, the shear strength obtain the highest, while with the increase of passes, for example, five and six passes, they have a negative effect on the properties of composite plate. We have added the relevant texts and figure to explain it. The added sentences and figure is shown as below:

Fig.2 The 1060Al/Q235 composite plate after M-FSP with different passes: (a) the 1060Al/Q235 composite plate after M-FSP; (a) single pass; (b) two passes; (c) three passes; (d) four passes

Fig.2 shows the 1060Al/Q235 composite plate after M-FSP with different passes. The repair zone for the 1060Al/Q235 composite plate after M-FSP with different passes are well-formed with no defect in it. However, with the increase of pass, especially for the passes are four, there shows obvious toe flashes after friction stir processing, as shown in Fig.2(e) with the red arrows. Beside this, according to the continuous axial force resulting in the toe flashes after M-FSP, the thickness of the aluminum plate becomes thinner gradually. Therefore, only four passes friction stir processing have been applied in this study.”

Round 2

Reviewer 1 Report

The paper can be accepted for publication.

Author Response

Question 1: The paper can be accepted for publication.

Response 1: Thank you for the reviewer for recognizing my work.

Reviewer 2 Report

The caption under the Fig. 2 is incorrect:

Fig.2 The 1060Al/Q235 composite plate after M-FSP with different passes: (a) the 1060Al/Q235 composite plate after M-FSP; (a) single pass; (b) two passes; (c) three passes; (d) four passes

It has to be:

Fig.2 The 1060Al/Q235 composite plate after M-FSP with different passes: (a) the 1060Al/Q235 composite plate after M-FSP; (b) single pass; (c) two passes; (d) three passes; (e) four passes

Author Response

Question 1: The caption under the Fig. 2 is incorrect:

Fig.2 The 1060Al/Q235 composite plate after M-FSP with different passes: (a) the 1060Al/Q235 composite plate after M-FSP; (a) single pass; (b) two passes; (c) three passes; (d) four passes

It has to be:

Fig.2 The 1060Al/Q235 composite plate after M-FSP with different passes: (a) the 1060Al/Q235 composite plate after M-FSP; (b) single pass; (c) two passes; (d) three passes; (e) four passes

Response 1: Thank you for the reviewer’s useful comment. We have corrected the mistake. The revised figure caption is shown as below:

“Fig.2 The 1060Al/Q235 composite plate after M-FSP with different passes: (a) the 1060Al/Q235 composite plate after M-FSP; (b) single pass; (c) two passes; (d) three passes; (e) four passes”

Reviewer 3 Report

  1. The wrong time was used, the past tense should be the same as in the preceding and following sentences: The axial force during M-FSP process is (should be 'was') 10kN. The FSW tool was made of W-Re alloy, and the FSW tool geometry consisted of a threaded conical pin, with three flats (7mm diameter) and a spiral (scroll) shoulder with a diameter of 25mm. The tilt angle for the FSW tool during welding is (should be 'was') 2°. The pin tool plunge depth is (should be 'was') 2.5mm.
  2. Definitive articles "the" should be removed from the captions of the drawings.

Author Response

Response to the Reviewer 3# Comment

Question 1: The wrong time was used, the past tense should be the same as in the preceding and following sentences: The axial force during M-FSP process is (should be 'was') 10kN. The FSW tool was made of W-Re alloy, and the FSW tool geometry consisted of a threaded conical pin, with three flats (7mm diameter) and a spiral (scroll) shoulder with a diameter of 25mm. The tilt angle for the FSW tool during welding is (should be 'was') 2°. The pin tool plunge depth is (should be 'was') 2.5mm.

Response 1: Thank you for the reviewer’s useful comment. We have corrected the mistakes. The revised sentences are shown as below:

“The axial force during M-FSP process was 10kN. The FSW tool was made of W-Re alloy, and the FSW tool geometry consisted of a threaded conical pin, with three flats (7mm diameter) and a spiral (scroll) shoulder with a diameter of 25mm. The tilt angle for the FSW tool during welding was 2°. The pin tool plunge depth was 2.5mm.”

Question 2: Definitive articles "the" should be removed from the captions of the drawings.

Response 2: Thank you for the reviewer’s useful comment. We have removed the definitive articles “the” from the captions of the drawings. The revised captions are shown as below:

“Table 1. Chemical compositions for 1060Al/Q235 composite plate (unit: wt.%)

Fig.2 1060Al/Q235 composite plate after M-FSP with different passes: (a) 1060Al/Q235 composite plate after M-FSP; (b) single pass; (c) two passes; (d) three passes; (e) four passes

Fig.3. Microstructures of the repair zone for 1060Al/Q235 composite plate after M-FSP with different passes: (a) single pass; (b) two passes; (c) three passes; (d) four passes

Table 2. Quantities and area of holes in the repair zone for 1060Al/Q235 composite plate after FSP with different passes

Fig.7. Interfaces of composite plate after FSP with different passes: (a) single pass; (b) two passes; (c) three passes; (d) four passes”
